# A Novel Chemical Gas Vapor Sensor Based on Photoluminescence Enhancement of Rugate Porous Silicon Filters

**DOI:** 10.3390/s20092722

**Published:** 2020-05-10

**Authors:** Zicheng Zhou, Honglae Sohn

**Affiliations:** 1Department of Chemistry, Chosun University, Gwangju 61452, Korea; zgzhou@126.com; 2School of Chemistry and Chemical Engineering, Cangzhou Normal University, Cangzhou 061001, China

**Keywords:** reflection, photoluminescence, porous silicon, rugate, organic vapor

## Abstract

In this study, an innovative rugate filter configuration porous silicon (PSi) with enhanced photoluminescence intensity was fabricated. The fabricated PSi exhibited dual optical properties with both sharp optical reflectivity and sharp photoluminescence (PL), and it was developed for use in organic vapor sensing. When the wavelength of the resonance peak from the rugate PSi filters is engineered to overlap with the emission band of the PL from the PSi quantum dots, the PL intensity is amplified, thus reducing the full width at half maximum (FWHM) of the PL band from 154 nm to 22 nm. The rugate PSi filters samples were fabricated by electrochemical etching of highly doped n-type silicon under illumination. The etching solution consisted of a 1:1 volume mixture of 48% hydrofluoric acid and absolute ethanol and photoluminescent rugate PSi filter was fabricated by etching while using a periodic sinusoidal wave current with 10 cycles. The obtained samples were characterized by scanning electron microscopy (SEM), and both reflection redshift and PL quenching were measured under exposure to organic vapors. The reflection redshift and PL quenching were both affected by the vapor pressure and dipole moment of the organic species.

## 1. Introduction

Since the discovery of its photo- and electro-luminescence characteristics [1,2], porous silicon (PSi) has been intensively researched for various applications, such as chemical [3] and biological sensors [4], drug delivery systems [5], and medical diagnostics [6]. PSi is an ideal candidate material for gas- and liquid-sensing applications due to its sponge structure with a high surface-to-volume ratio, various direction of pores, and pore diameters that can be altered by changing the production conditions [7,8]. The sensing techniques that have been mainly investigated to achieve signal transduction have been based on capacitance [9], resistivity [10], reflectivity [11], and photo-luminescence (PL) [12]. However, optical measurements for detection applications have demonstrated high sensitivity and potentially greater capability for the identification of specific adsorbates.

PL emission is most likely due to the quantum confinement effects within silicon quantum dots, to the best of our knowledge. The enhancement of PL is critical for improving the detection quality. It was reported in the literature that increasing radiative recombination via direct band-gap transitions and reducing the phonon-assisted indirect band-gap transitions can significantly enhance the photoluminescence [13]. Previous studies led by our group have demonstrated the successful fabrication of the DBR multilayer porous silicon devices exhibiting sharp PL peaks that arise from Bragg resonance in the visible range [14]. DBR devices are fabricated by alternating the square wave current between the high and low current values, leading to layers with low and high refractive indices, respectively. However, if the current is modulated gradually, PSi with a smooth variation in the refractive indices along with etching direction can be achieved, obtaining the so-called rugate PSi. In this work, luminescent rugate PSi with narrow and strong PL was prepared while using a highly doped n-type silicon wafer under illumination. We designed the reflection peak that is exclusively reflected in the range of the rugate PL wavelength position and constructively propagates with the overlapping emission domain of the optical Si quantum dots, thus amplifying the PL intensity. The present work is the first demonstration of the use of this approach for PL enhancement, to the best of our knowledge.

The PL efficiency of PSi relies on the interior diverse structures and the versatile surface properties [15]. Organic vapors have been detected by PL quenching due to the differences in the physicochemical properties between the organic molecules [16]. Here, we report a method for the preparation of photoluminescent rugate PSi exhibiting both well reflection and PL based on highly doped n-type silicon wafers. The PL that was enhanced by constructive propagation with reflection from porous silicon was used for the detection of the vapor pressure of organic solvents.

## 2. Materials and Methods

### 2.1. Preparation of Rugate PSi and Monolayer PSi

Rugate PSi was prepared by an electrochemical anodization of an n^++^-type (0.001~0.003 Ω·cm) single-side polished <100> oriented silicon wafer (Prime grade, Siltronix Inc., Archamps, France). Anodization was performed while using a Teflon cell with Pt ring as the counter electrode at room temperature in a mixed solution of 50% volumetric fraction of aqueous HF (48 wt%, Aldrich Chemicals, Sigma-Aldrich, Steinheim, Germany) and absolute ethanol (ACS reagent, Aldrich Chemicals, Sigma-Aldrich, Steinheim, Germany) with the volume ratio of 1:1. The applied current density was controlled using a Keithley 2420 (Keithley Instruments Inc., Cleveland, OH, USA). The pseudo sinusoidal waveform involves an individual sine component, which is represented as:*y* = *A*sin(*kt*) + *B*(1)
where *y* represents a temporal sine wave with amplitude *A*, frequency *k*, time *t*, and an applied current density *B*. The values for *A*, *k*, *t*, and *B* were set to 60.55, 1.02, 1000, and 110, respectively. Gradually varied etching that was controlled by sinusoidal current was performed under the illumination with a 300-W tungsten filament bulb for the entire duration of etching. For comparison, a monolayer PSi exhibiting broad PL was prepared by applying 200 mA cm^−2^ for 300 s under the illumination with the same bulb during etching. All of the samples were then rinsed several times by ethanol and then dried under argon atmosphere prior to use.

### 2.2. Instruments and Data Acquisitions

Steady-state PL spectra were measured using an Ocean Optics S2000 spectrometer that was fitted with a fiber optic probe. PL spectra were recorded with an excitation wavelength of 460 nm. Percent quenching values are reported as (*I*_0_ − *I*)/*I*_0_, where *I*_0_ is the initial PL intensity and *I* is the PL intensity after quenching. The reflectance spectra were obtained with a white light and recorded with a CCD detector in the wavelength range of 400~1200 nm. The morphologies of rugate PSi and monolayer PSi samples were obtained by cold field emission scanning electron microscopy (FE-SEM, S-4800, Hitachi, Tokyo, Japan). 

### 2.3. Detection of Organic Vapor 

The effectiveness of the amplified PL in regards to detection performance by rugate PSi was examined by sensing organic solvents. The results of organic vapor detection performance by rugate PSi were compared with monolayer PSi. The rugate PSi were placed in an exposure chamber that was fitted with an optical window, and toluene, hexane, and chloroform were used to investigate the adsorption behavior. Argon gas was used as the carrier gas for the volatile organic compound vapors, and the chemical vapor concentration was adjusted by the flow meters. The influence of different organic species vapors, namely, toluene, hexane, and chloroform, on the PL spectra of rugate PSi was studied. All of the sensing measurements were carried out at room temperature. Figure 1 showed a schematic diagram for the gas sensing system. We measured the PL of rugate PSi in the presence of different organic species with various vapor concentrations at the same carrier gas flow. The results showed that PL spectra are affected by the volatile organic vapor, and this effect can be exploited to fabricate organic humidity sensors based on rugate PSi filters. Optical reflectivity spectra were measured while using a tungsten–halogen lamp and a CCD spectrometer that was fitted with a fiber optic input.

## 3. Results and Discussion

Rugate PSi exhibiting narrow peaks for both reflectance and PL was successfully fabricated by applying gradually varied current waveform under illumination with a 300-W tungsten filament bulb for the duration of the etching. Figure 2 shows photographs of rugate PSi and monolayer PSi samples under visible and UV light. Plotted together, the reflection and PL spectra (Figure 3) show that the position of the PL peak from rugate PSi coincides with the reflection peak of rugate PSi at 663 nm with a full width at half maximum (FWHM) of 22 nm. Thus, the optical peak was enhanced because the photoluminescence constructively propagated with reflectance. In contrast, the optical spectra of monolayer PSi showed a broad PL spectrum with a FWHM of 154 nm.

The rugate PSi having gradually modulated refractive indices with sinusoidal porosity exhibits a narrow and high reflection band. However, monolayer PSi having constant refractive index with cylindrical pore exhibits a lack of reflection features. For photoluminescence, the PL of monolayer and rugate PSi is due to the quantum confinement of silicon quantum dots (QDs) on the surface of PSi. Figure 4A,B display schematic illustrations for rugate and monolayer PSi, respectively. Figure 4C shows that the emission spectrum of Si QDs in rugate PSi was substantially narrowed and amplified after passing through the special and repeated multilayers of rugate structure. However, the emission spectrum of monolayer PSi showed a broad PL band due to the lack of rugate filter. 

Figure 5 shows the SEM images of the surface and cross-sectional views of the rugate and monolayer PSi. Figure 5A showed that the rugate PSi that was prepared with 10 cycles exhibited a flat surface with the pore sizes in the 10–20 nm range. The total thickness of the rugate and monolayer PSi were approximately 88.5 and 82.5 μm, respectively. Rugate PSi, as shown in Figure 5B, is composed of a regularly changing special stripe structure that is gradually varied with the sinusoidal porous layers. This gradually varied porous layer is consistent with the alternating sinusoidal etching current. By contrast, a cylindrical pore structure for the monolayer PSi that was fabricated using a constant etching current was generated (Figure 5C,D).

Organic vapors were adsorbed on the pore inside surface and then capillary condensation occurred, leading to the shift of the reflectivity of the rugate PSi to longer wavelengths due to the increase in the refractive indices of the porous medium, as shown in Figure 6. The observed redshift was triggered by the increased average refractive indices of the porous medium was induced by the partial substitution of air by the organic liquid phase in the multilayer pores of rugate PSi due to the capillary condensation effect.

The obtained reflection shifts for these organic substances are reported in Figure 6 as a function of their corresponding refractive indices. An examination of the results showed that different organic species shown in Figure 7 result in different resonance shifts. However, even though the redshift was induced by the increase of a refractive index, it did not have a definite relationship with the refractive index of the analyte. Table 1 shows the refractive index data of the above analytes. For substances having close values of refractive index, namely toluene and chloroform, the reflection peak shift can be distinguished well. In addition, higher refractive indices do not necessarily translate to more prominent shifts; for example, toluene has a relatively higher refractive index, but it was found to exhibit the smallest reflectivity redshifts. These results are very inconsistent with the reports that the reflection redshift exhibits a positive linear dependence on the refractive index [17,18,19]. The discrepancy can be explained by the fact that the redshift is dominated by not only the refractive index of organic species, but also the filling fraction of the stratified silicon pores, due to capillary condensation. Our previous work found that the PSi reflectivity shift range was directly dependent on the vapor pressure of organic molecules [20]. The vapor pressure increases gradually for toluene, hexane, and chloroform, as shown in Table 1. Based on the above results, the extent of the reflectivity redshift was affected by the vapor pressure of the organic solvents. The organic molecules are adsorbed on the surface of the porous silicon and then undergo capillary condensation that changes the refractive index of the rugate PSi filters. The different extent of the reflection redshift generated due to the different concentration of organic solvents in carrier gas is due to the different filling fraction of the organic compound in the silicon pores.

An examination of Figure 2 indicates that the PL quenching of rugate PSi filters is due to the organic solvent molecules that are physisorbed on the surface of the luminescent chromophore and then undergo capillary condensation inside the pores. We predict that rugate PSi filters will exhibit greater sensitivity than monolayer PSi with a broad PL peak based on the observed sharp PL decrease in the presence of organic solvent vapor. An organic vapor sensor based on rugate PSi filters was fabricated and tested for different chemical vapor pressures of toluene, hexane, and chloroform in order to investigate the PL quenching sensitivity behaviors of the organic vapor. Figure 8A–C showed a shift of the reflectivity of the rugate PSi to longer wavelengths due to the increase in the refractive indices of the porous medium. The steady-state PL spectra shown in Figure 8D–F displayed PL quenching under the exposure to the vapor pressure of the above three organic species. Organic vapor (carrier gas flow was 1 SLM) was injected into the surface of rugate PSi sample, resulting in reversible PL quenching. The PL intensity of the rugate PSi in vacuum can be recovered to that of the original spectrum. The PL intensities for different organic species vapors were decreased when compared to the initial PL. However, the PL quenching magnitudes of the rugate PSi filters varied greatly and were 6.7%, 11.4%, and 52.7% within 20 s for toluene, hexane, and chloroform, respectively. However, for monolayer PSi shown in Figure 8G–I, the corresponding values were 5.1%, 9.5%, and 42.9%, respectively. These results indicated that the sharp PL peak of the rugate PSi filters is more sensitive than the broad PL of the monolayer PSi. The reflection detection showed that reflectivity peak redshifted by 27, 58, and 53 nm within 20 s. Even though the vapor pressure of chloroform was higher than that of hexane, the reflection redshift for chloroform was smaller than that for hexane. The PL wavelength redshifts were 17, 33, and 28 nm for toluene, hexane, and chloroform, respectively, within 20 s during sensing. The PL redshift of chloroform was also smaller than that of hexane. This result is due to the much smaller polarity of hexane when compared to chloroform. Hexane adsorbed more easily on the pore surface of PSi since the fresh porous silicon surface was hydrophobic. 

Figure 3 shows that the PL was enhanced by reflection, and the PL and reflectance peaks overlapped at the same wavelength position. The reflection and PL both redshifted during the organic solvents sensing, which might be due to the increase of refractive indices. However, the extent of the PL redshift was smaller than that of the reflection redshift, illustrating that the PL redshift is induced by the reflection redshift. This conclusion is also supported by the similar behavior of the reflection redshift and PL redshift during the first five seconds larger than the subsequent interval time of five seconds as shown in the upper half part of Figure 8, respectively.

In Figure 8, the PL of the monolayer PSi was blue-shifted by 30 nm when it was exposed to chloroform vapor, but to a lesser extent for hexane (4 nm) and toluene (2 nm) vapors. Sailor reported that the PL wavelength was blue-shifted from 670 nm to 630 nm when PSi was exposed to tetrahydrofuran vapor [21]. These results may be due to the faster PL emission decay of the monolayer PSi at the longer wavelengths, which was determined by different excited state lifetime for emission (τ) at different wavelength. Figure 9 showed different quenching rates of excited emission at 650, 700, and 750 nm for monolayer PSi under the exposure of chloroform vapor. 

The left figure in Figure 10 shows a three-dimensional plot for the identification of the analyte while using the relationship between the shifts in the PL wavelength, PL quenching, and the vapor pressure of three different analytes. The rugate PSi was exposed three times to each analyte and the three spheres that represent the results of the measurements are shown in the same color and connected by lines crowd together, forming a cluster. A smaller area of the cluster indicates better reproducibility and reliability. The right figure in Figure 10 shows the three-dimensional plot used for the identification of the functional relationships between the quenching percentage of the normalized PL area, PL wavelength shift, and the sensing time of the three different analytes. The different directions of testing curves validate that toluene, hexane, and chloroform can be clearly distinguished, which means that the volatile organic compounds detection using rugate PSi is specific and effective.

Previous work in our group has demonstrated that the dynamic Stern–Volmer quenching model shows linear relationships between PL quenching and quencher concentration for different analytes [21]. However, the Stern–Volmer constants for the different analytes were not identical. These results indicated that the PL quenching depended on not only the vapor pressure of the analytes, but also the functionalities of organic molecules. Additionally, Nayef et al. reported [22] that PL quenching is related to the dipole moment of organic molecules. This finding can be interpreted as being due to the stabilization of the PSi surface trap by the alignment of the dipoles of the organic molecules. In the present study, the observed PL quenching of chloroform was much larger than that for hexane, despite their similar vapor pressure, which should be attributed to the larger dipole moment of chloroform. However, the dipole moment of toluene is larger than that of hexane, but its PL quenching was smaller than hexane. This result is consistent with our previous reports that PL quenching depends on the vapor pressure. The vapor pressure and polarity of the analytes affected the reflectance results and, in turn, the polarity of the analyte depends on the dipole moment of organic molecules. Thus, both the analyte vapor pressure and dipole moment are important factors for PL quenching used as the basis for volatile organic compounds detection.

## 4. Conclusions

A preparation of rugate PSi filters exhibiting enhanced PL through the constructive overlap with reflectivity was reported. The PL changes were used for the detection of different volatile organic compounds with different vapor pressures and dipole moments. The vapor pressure and dipole moment of the organic compounds both affected the reflectivity redshift and PL quenching of rugate PSi. It is concluded that rugate PSi filters can be considered for use in chemical vapor sensors that are based on the reflectivity redshift and PL quenching. 

## Figures and Tables

**Figure 1 sensors-20-02722-f001:**
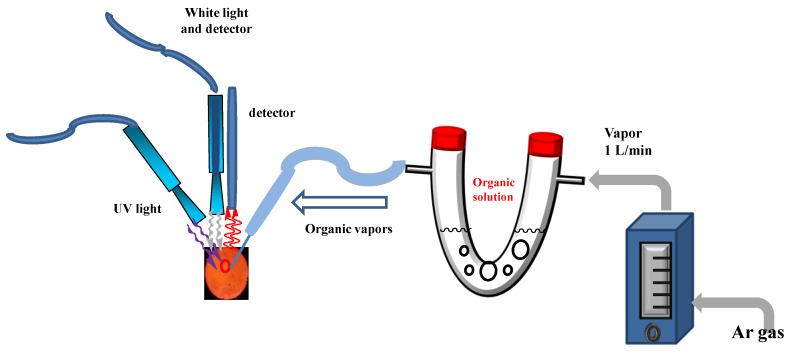
Schematic diagram of the gas measuring system.

**Figure 2 sensors-20-02722-f002:**
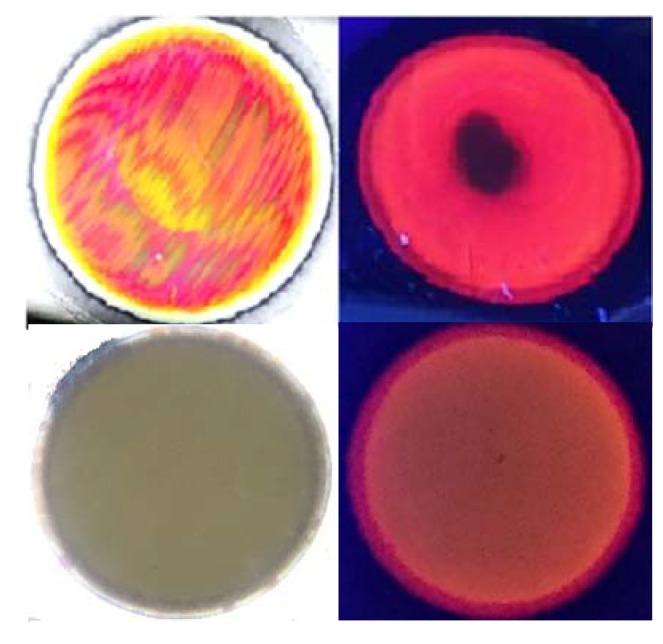
Photographs of rugate porous silicon (PSi) (upper) and monolayer PSi (below) samples under visible light (left) and ultraviolet light (right). Dark area of the red luminescent region in rugate PSi represents the quenching of photoluminescence (PL) with a drop of hexane.

**Figure 3 sensors-20-02722-f003:**
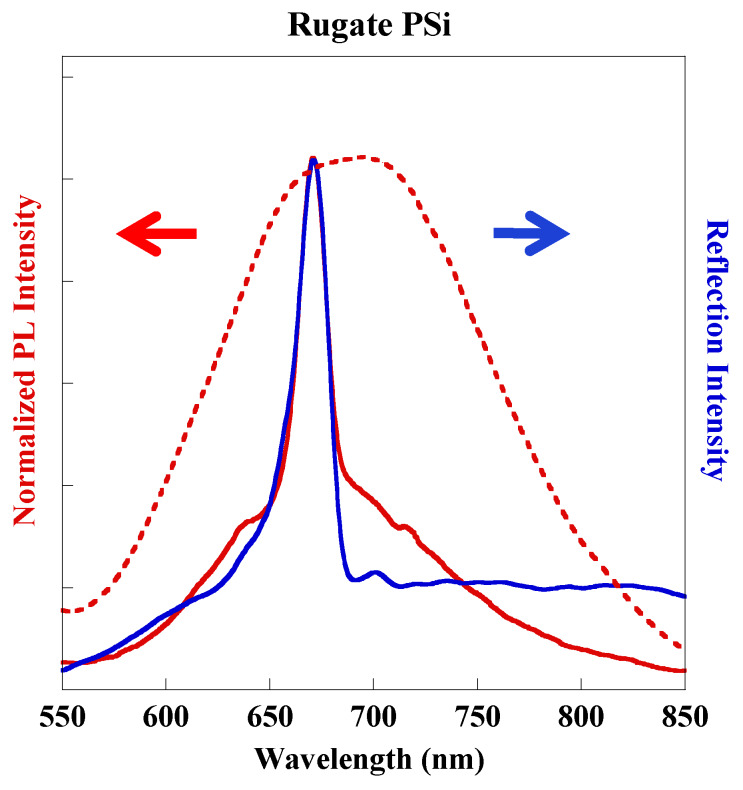
PL (red line) and reflectance (blue line) spectra of the prepared rugate PSi and PL (red dot) spectrum of the monolayer PSi.

**Figure 4 sensors-20-02722-f004:**
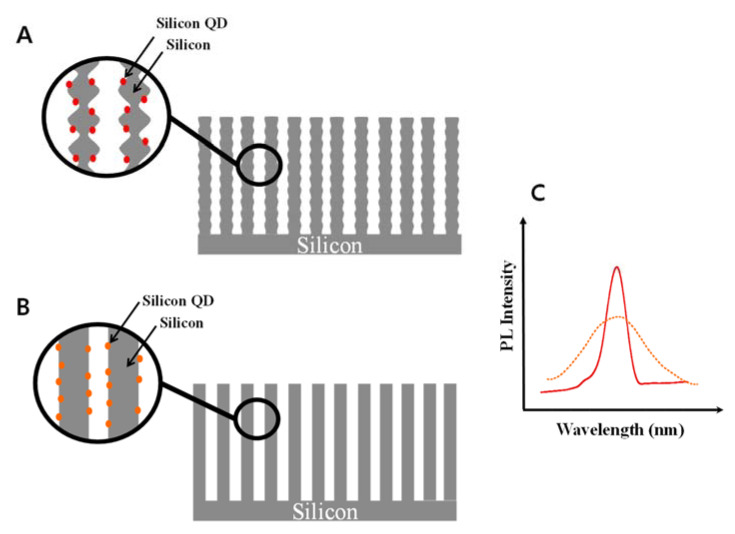
Schematic illustration of (**A**) rugate PSi and (**B**) monolayer PSi. PL spectra for (**C**) rugate PSi (solid line) and monolayer PSi (dotted line).

**Figure 5 sensors-20-02722-f005:**
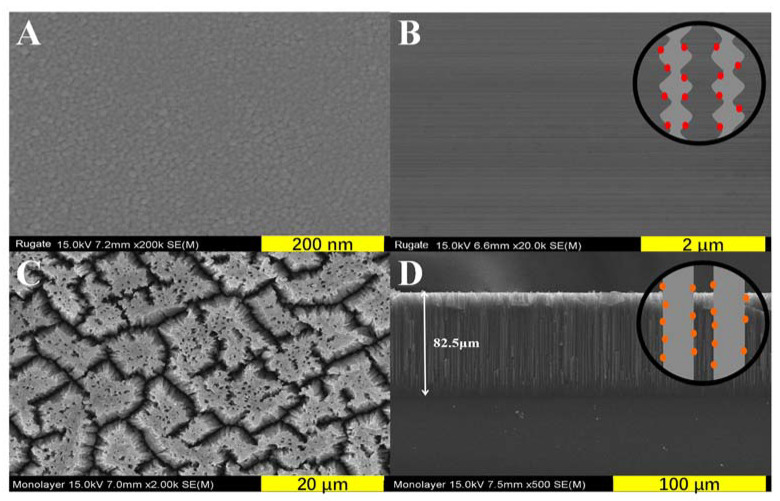
Surface and cross-section SEM images of rugate PSi (**A**,**B**) and monolayer PSi (**C**,**D**).

**Figure 6 sensors-20-02722-f006:**
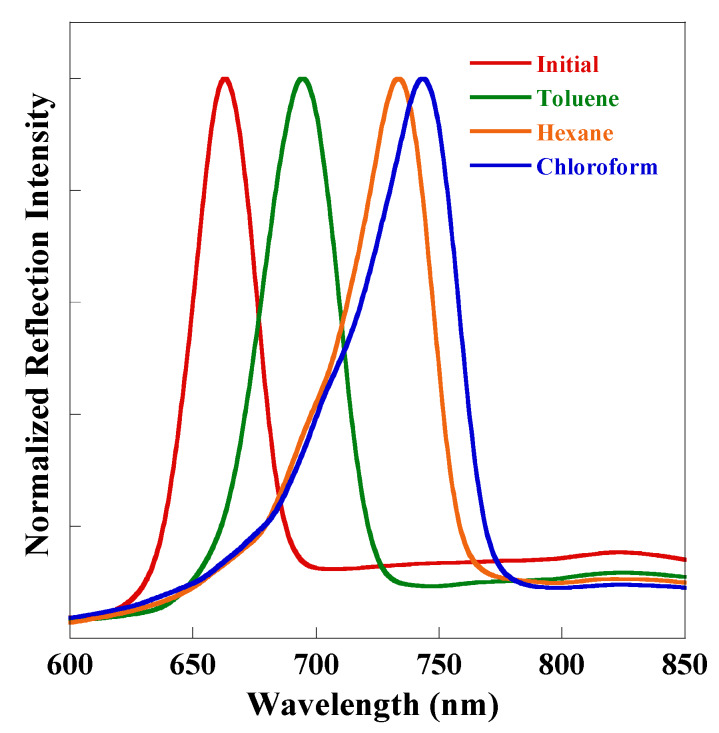
Reflection spectra of rugate PSi filters (red: λ = 663 nm) under a flux of toluene (green, λ = 695 nm, Δλ = 32 nm), hexane (orange, λ = 734 nm, Δλ = 71 nm), and chloroform (blue, λ = 744 nm, Δλ = 81 nm).

**Figure 7 sensors-20-02722-f007:**
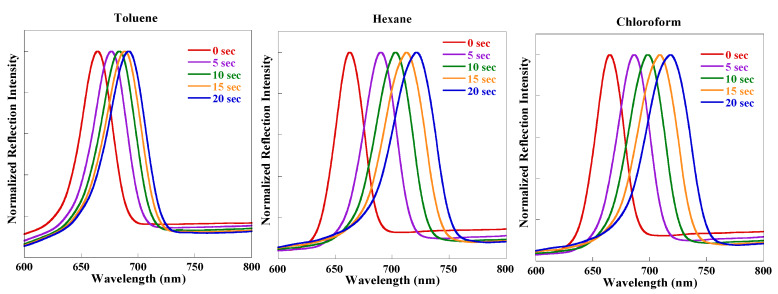
Reflection spectra of rugate PSi within 20 s under the same argon flow (1 SLM) for three different organic solvents.

**Figure 8 sensors-20-02722-f008:**
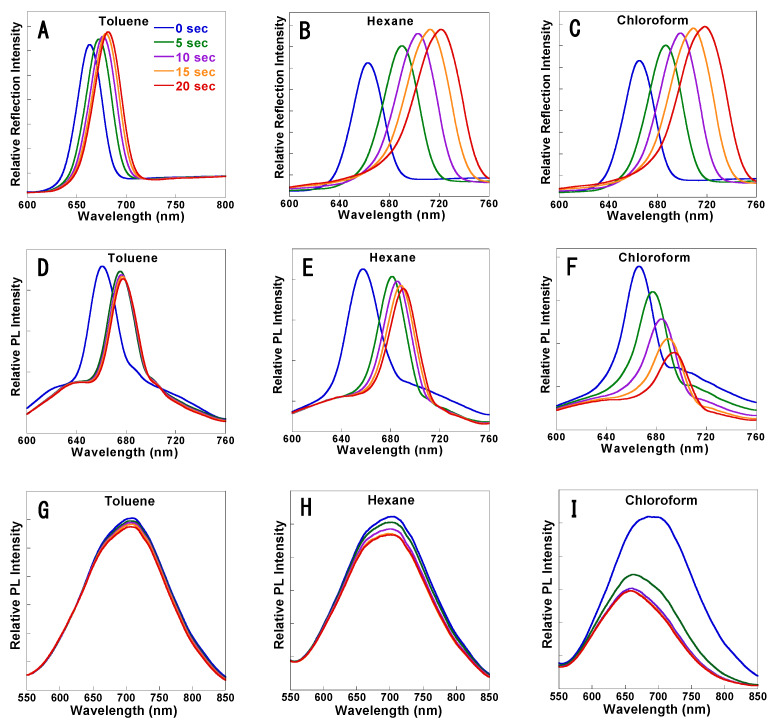
Reflection spectra (**A**–**C**) and Time-resolved PL (**D**–**F**) of rugate PSi and monolayer PSi (**G**–**I**, only PL) under the exposure of different organic vapors with identical argon flow rate (1 SLM) for 20 s.

**Figure 9 sensors-20-02722-f009:**
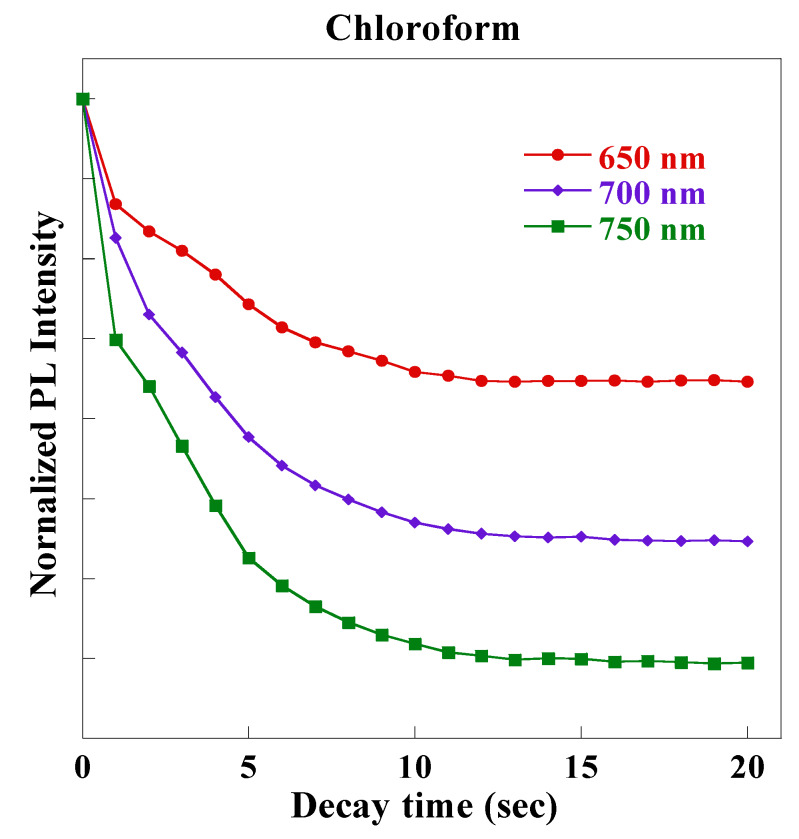
Time-resolved PL emission decays of monolayer PSi at different wavelengths under chloroform vapor exposure.

**Figure 10 sensors-20-02722-f010:**
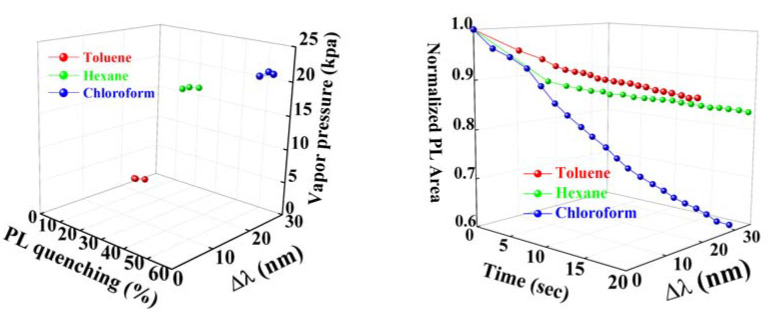
LEFT: three-dimensional plot showing the relationship between the wavelength shift of the PL and PL quenching as a function of different vapor pressures of the three analytes. RIGHT: three-dimensional plot showing the relationship between the normalized PL area, the PL wavelength shift as a function of different vapor pressures of the three analytes.

**Table 1 sensors-20-02722-t001:** Physical parameters of methanol, hexane, and chloroform.

Organic Species	Toluene	Hexane	Chloroform
Refractive index	1.497	1.354	1.446
Dipole moment/D	0.36	0.08	1.15
Vapor Pressure/kPa	2.8	16.2	21.2

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
