# Peer review of "A Novel Chemical Gas Vapor Sensor Based on Photoluminescence Enhancement of Rugate Porous Silicon Filters"

_sensors, 2020, doi:10.3390/s20092722_

Round 1

Reviewer 1 Report

1.Why does PL spectra have blueshift when chloroform was adsorbed on monolayer PSi in figure 7? This picture is different from others , but the paper doesn’t explain why.

2.In the figure 7, the description states the PL attenuation of chloroform is 16.4%. But the words in 248 lines in the paper states the PL attenuation of that is 42.9%. It is a little conflicting.

3. The picture B in figure 3 overlay each other. This can be adjusted.

4.Can you show your experiment details? We don’t know that the luminous intensity.

5. The peak of rugate Psi reflection spectra in Figure 5 (663nm) is different from the data in line 103(671nm).

6.The paper indicated that the advantage of rugate Psi is PL peak coincides with the reflection peak (Figure 2). But the organic gas adsorption experiment doesn’t show the resonance of two waves. I think resonant wave is easier to see the advantages of this material.

Author Response

(1) Why does PL spectra have blueshift when chloroform was adsorbed on monolayer PSi in figure 7? This picture is different from others , but the paper doesn’t explain why.

Reply to comment (1) : The reason has been explained in page 8 and figure 9 has been newly added to support an explanation.

Actually, PL spectra of monolayer PSi were all blue-shifted in different extent when it was sensing toluene, hexane, and chloroform. These shifts were due to the faster PL emission decay of the monolayer PSi at the longer wavelengths, which was determined by different excited state lifetime for emission (τ) at different wavelength. Figure 9 showed different quenching rates of excited emission at 650, 700, and 750 nm for monolayer PSi under the exposure of chloroform vapor.

(2) In the figure 7, the description states the PL attenuation of chloroform is 16.4%. But the words in 248 lines in the paper states the PL attenuation of that is 42.9%. It is a little conflicting.

Reply to comment (2) :There was a mistake in legend of figure 7 ( figure 8 in the revised version). It has been corrected.

(3) The picture B in figure 3 overlay each other. This can be adjusted.

Reply to comment (3) : Figure 3 ( figure 4 in the revised version) has been corrected according to the referee’s suggestion.

(4) Can you show your experiment details? We don’t know that the luminous intensity.

Reply to comment (4) : The detailed preparation of rugate and monolayer PSi described in chapter 2.1, and the experimental procedure of organic vapor detection have been replenished as detailed as possible. The PL photograph of monolayer PSi under ultraviolet light has been replenished to see the luminous intensity by naked eyes.

(5) The peak of rugate PSi reflection spectra in Figure 5 (663nm) is different from the data in line 103(671nm).

Reply to comment (5) : The data of PL and reflection coincided at 663nm which has been corrected.
.

(6) The paper indicated that the advantage of rugate PSi is PL peak coincides with the reflection peak (Figure 2). But the organic gas adsorption experiment doesn’t show the resonance of two waves. I think resonant wave is easier to see the advantages of this material.

Reply to comment (6) : According to the Referee’s suggestion, as shown in figure 7, two resonance waves of PL quenching and reflection blued-shift have been shown together during organic gas sensing . It is a good idea which could be compared and could be evidenced the advantages of this material.

Reviewer 2 Report

The paper should include a scheme of gas measuring system (in the form of a drawing or block type diagram).

The paper should include the picture of the structure, the readers have to read well to understand what the structure looks like.

Figure 3B: the Y axis captin is cut off.

Figure 4: this figure should be corrected, not just the SEM imagges, only the way they are presented. The scale line is illegible, in 4 cases out of 5 it is too small. Why A1: section is so big compared to the other images? You wrote that some images were made in "section" or "top" but without a structure diagram it is hard to guess from which part it was taken.

Figure 5: there should be a legend.

Figure 8: A 3D graph is an interesting form of presenting data, but you must remember that it must be legible. The axis captions, especially in the figure on the right, are illegible.

line 71: there should be ":" instead "." after "as"

Author Response

The following points should be addressed:
(1) The paper should include a scheme of gas measuring system (in the form of a drawing or block type diagram).

Reply to comment 1 : Figure 1 has been replenished according to the Referee’s suggestion. It is much better to know how to achieve organic gas sensing.

(2) The paper should include the picture of the structure, the readers have to read well to understand what the structure looks like.

Reply to comment 2 : Figure 4 has been changed for better understanding of structure of rugate PSi.

(3) Figure 3B: the Y axis caption is cut off.

Reply to comment 3 : Figure 3 has been modified according to the Referee’s suggestion and placed in Figure 4.

(4) Figure 4: this figure should be corrected, not just the SEM imagges, only the way they are presented. The scale line is illegible, in 4 cases out of 5 it is too small. Why A1: section is so big compared to the other images? You wrote that some images were made in "section" or "top" but without a structure diagram it is hard to guess from which part it was taken.

Reply to comment (4) : Figure 4 has been modified in Figure 5 according to the Referee’s suggestion.

(5) Figure 5: there should be a legend.

Reply to comment (5) : Figure 5 has been fixed and placed in Figure 6.

(6) Figure 8: A 3D graph is an interesting form of presenting data, but you must remember that it must be legible. The axis captions, especially in the figure on the right, are illegible.

Reply to comment (6) : The size and orientation of axis captions have been changed.

(7) line 71: there should be ":" instead "." after "as".

Reply to comment (7) : The mistake of punctuation has been corrected.

Round 2

Reviewer 2 Report

I recommend the article for publication in the Sensors.